# Proteomics Answers Which Yeast Genes Are Specific for Baking, Brewing, and Ethanol Production

**DOI:** 10.3390/bioengineering7040147

**Published:** 2020-11-18

**Authors:** Svetlana Davydenko, Tatiana Meledina, Alexey Mittenberg, Sergey Shabelnikov, Maksim Vonsky, Artyom Morozov

**Affiliations:** 1Innovation & Research Department, Baltika Breweries—Part of the Carlsberg Group, 6-th Verkhnij ln. 3, 194292 St. Petersburg, Russia; davydenko@baltika.com; 2Faculty of Biotechnologies (BioTech), ITMO University, Lomonosova st. 9, 191002 St. Petersburg, Russia; tatiana.meledina@yandex.ru; 3Proteomics and Mass Spectrometry Group, Cell Technologies Center, Institute of Cytology, Russian Academy of Sciences, Tikhoretsky av. 4, 194064 St. Petersburg, Russia; a.mittenberg@gmail.com (A.M.); sergey_shabelnikov@incras.ru (S.S.); 4Department of State Standards and Reference Materials in the Area of Bioanalytical and Medical Measurements, D.I. Mendeleyev Institute for Metrology VNIIM, Moskovsky pr. 19, 190005 St. Petersburg, Russia; m.s.vonsky@vniim.ru

**Keywords:** yeast, domestication, protein expression, brewing, baking, ethanol production, industrial strains

## Abstract

Yeast strains are convenient models for studying domestication processes. The ability of yeast to ferment carbon sources from various substrates and to produce ethanol and carbon dioxide is the core of brewing, winemaking, and ethanol production technologies. The present study reveals the differences among yeast strains used in various industries. To understand this, we performed a proteomic study of industrial *Saccharomyces*
*cerevisiae* strains followed by a comparative analysis of available yeast genetic data. Individual protein expression levels in domesticated strains from different industries indicated modulation resulting from response to technological environments. The innovative nature of this research was the discovery of genes overexpressed in yeast strains adapted to brewing, baking, and ethanol production, typical genes for specific domestication were found. We discovered a gene set typical for brewer’s yeast strains. Baker’s yeast had a specific gene adapted to osmotic stress. Toxic stress was typical for yeast used for ethanol production. The data obtained can be applied for targeted improvement of industrial strains.

## 1. Introduction

Many applications of wild yeast species have been found for human needs. During the process of domestication, microbes have acquired the ability to efficiently assimilate certain nutrients, cope with a variety of industry-specific stress factors, and produce targeted compounds, resulting in the emergence of genetically and phenotypically different strains. The genome of *S. cerevisiae* has undergone evolutionary formation over thousands of years under the influence of domestication events. Humans have interfered with this natural process of evolution by creating artificial niches and selecting suitable phenotypes.

*S. cerevisiae* is used in the production of food, beverages, and biofuels, and as a cell factory for the production of pharmaceuticals and other biochemical compounds. Therefore, the role of yeast in human life has stimulated scientific interest, and yeast strains have become important model organisms for laboratory research of practical significance for industry and medicine [1,2,3,4,5]. *S. cerevisiae* genome was the first eukaryotic genome to be sequenced [6]. Currently, the *S. cerevisiae* genome database is publicly available [7].

One of the most important events in the evolution of *S. cerevisiae* has been the entire genome duplication. This provided *S. cerevisiae* with a number of advantages, including surviving in sugar-rich ecological niches, high fermentation capacity, the Crabtree effect, and the MAC (make-accumulate-consume) strategy. These advantages have allowed for the direction of the metabolic flow from simple sugars to ethanol, reprogramming, and loss of regulatory elements of genes involved in respiration [8].

*S. cerevisiae* yeast strains used for beer, bread, and wine production are genetically and phenotypically different from natural isolates. The complexity of studying the origin of domesticated strains is associated with human migration and the history of crossing with wild yeast natural isolates. Fay J. C. et al., in 2019, studied the population genetic history of brewer’s yeast strains and found that ale strains and some allotetraploid lager strains, derived from a mixture of populations, were closely related to European grape wine yeast strains and Asian rice wine yeast strains. Similar to baker’s strains, brewer’s strains are polyploid. This prevents interbreeding and ensures their isolation from other populations [9].

Currently, bioarchaeology, which searches for modern industrial yeast strain ancestors, has attracted significant interest [10]. Hybridization among different species often leads to non-viable or infertile offspring, but there are examples of such interspecific hybrids in animals, plants, and microorganisms used by humans. Brewer’s yeast is an example of such hybridization. In 2019, Gallone B. et al. analyzed a large set of interspecific yeast hybrids isolated from the brewing environment and showed that hybrids between *Saccharomyces* species arose and diversified in industrial conditions by combining the key features of each parent species. In addition, the post-hybridization evolution within each hybrid line has reflected specialization and adaptation to certain styles of beer [11]. Thus, interspecific hybridization provides an important evolutionary pathway for rapid adaptation to new conditions. Yeast strains have been studied extensively, and they are convenient models for studying domestication processes. The ability of yeast to ferment carbon sources located in various substrates and to produce ethanol and carbon dioxide provides the basis for the development of baking, brewing, winemaking, and ethanol production. The yeast cell cycle features, mating type, as well as haplo- and diplophase have contributed to advancements in genetic research. The high survival rate of cells with chromosomal rearrangements and additional chromosomes has ensured survival of aneuploid and polyploid cells and selection of new industrial strains. Sequence analysis have been used to compare different yeast isolates, to trace the genetic relationship and evolutionary changes in their genomes. Figure 1 shows the main mechanisms of yeast genomes evolution, i.e., horizontal gene transfer, changes in the gene copies number, chromosomal rearrangements, hybrid genome breakdown, intraspecific hybridization. These mechanisms have allowed industrial yeast strains to acquire such unique properties as low-temperature fermentation, transport of oligopeptides into the cell, reduction of beer aroma defects, and resistance to sulfite and copper ions.

Attempts to create new yeast strains for brewing by hybridizing of ancestral parent species (*S. cerevisiae* and *Saccharomyces eubayanus*) have led to the creation of strains with defects in the aromatic profile of beer. Unfortunately, in addition to the desired properties, these new yeasts have also inherited undesirable characteristics. The most notable has been phenolic off-flavors (POF) production. These flaws have prevented the use of such yeast strains directly in industrial production processes. However, the use of modern technologies for gene expression modification offers a new tool for the targeted phenotype correction of *Saccharomyces sp.* [12,13] and improving undesirable properties of industrial strains. For instance, genome editing technology based on the associated CRISPR-Cas9 has created a mutation in the ferulic acid decarboxylase *FDC1* gene. It is involved in the decarboxylation of aromatic carboxylic acids such as phenylacrylic (cinnamic) acid, ferulic acid, and coumaric acid, as well in the formation of the corresponding vinyl derivatives. POF^−^ strains obtained with this technology have a high potential for industrial applications [14].

In addition, CRISPR-based technology has successfully reduced urea production in wine yeast [15] and introduced a hop monoterpenes biosynthesis pathway into the brewer’s yeast genome, enabling production of certain hop flavors [16]. In 2017, de Vries A.R.G. and colleagues demonstrated the efficiency of specific phenotypes modifications, such as esters production, by knocking out responsible genes [17]. Although CRISPR-based technologies have been demonstrated to have wide potential, their practical application has been limited by national legislations [18].

The need to obtain highly productive strains is an urgent demand for biotechnology industries. However, the task of constructing such strains is complicated due to the lack of selective phenotypes. Specific targets must be preliminarily identified to improve further selection approaches. To determine perspective target genes, we performed a proteomics study of yeast strains related to different fermentation processes.

Our investigation aimed to reveal the differences among yeasts used in different industries. This information could contribute to the creation of new industrial strains with improved properties by applying new technologies that are able to change the activity of certain genes. To solve this task, we used the proteomics and genetic databases of yeast *S. cerevisiae.*

## 2. Materials and Methods

As the nutrient media, YPD is a complete culture medium for yeast growth and contains 1% yeast extract, 2% peptone, 2% glucose, and distilled water; solid medium contains 2% agar-agar [19].

The following strains of *S. cerevisiae* yeast were used in the study: LV7 and Ethanol Red (ER) (collection of ITMO University) and Y-3194 (collection of Baltika Breweries [20] and VKPM [21]). These yeast strains were selected for the study due to their widespread use in different fermentation processes in beer, bread, and distillation industries.

Yeast cultures were grown on a solid YPD medium for 24 h at 27 °C for stationary phase, then inoculated at an initial concentration of 10^6^ cells/mL in flasks with 100 mL of liquid YPD and grown under constant agitation at 150 rpm, for 18 h, at 27 °C. Yeast cells of all strains were cultivated simultaneously with the same YPD batch.

For two-dimensional electrophoresis of proteasome-associated proteins, isoelectric focusing was performed in IPG (immobilized pI gradient) strips (11 cm long, pH 3–11 NL (non-linear)) in IPG Phor 3 IEF (isoelectric focusing) system (GE Healthcare) [22] followed by SDS-PAGE separation according to Laemmli U.K. at 1970 with some modifications [23]. The proteins were visualized by Coomassie brilliant blue G-250 staining.

For peptide fractionation, peptides were separated with a Jupiter Proteo C12 reversed-phase column (1 × 50 mm, 4 μm, 90 Å; Phenomenex, CA, USA) on a microflow high-performance liquid chromatography (HPLC) (system (MiLiChrom A-02, EcoNova, Novosibirsk, Russia). A sample volume of 50 μL was injected and separated using a linear gradient of 10–35% B over 54 min followed by 35–90% B for 6 min, at a flow rate of 50 μL·min^−1^. The mobile phases used were A, 0.125% (*v*/*v*) trifluoroacetic acid (TFA) in water and B, 0.125% (*v*/*v*) TFA in acetonitrile. The column was maintained at 45 °C. The effluent from the HPLC column was mixed with alpha-cyano-4-hydroxycinnamic acid (CHCA) matrix (12 mg·mL^−1^ in 95% acetonitrile) at a flow rate of 15 μL·min^−1^ via a micro tee. A micro-fraction collector was used to deposit a total of 912 fractions of 0.5 μL in a 24 × 38 array on an LC-MALDI plate (SCIEX, Darmstadt, Germany). The column was washed with a saw-tooth gradient (15–80–15% B for 4 min and 2 min, respectively, repeated eight times) and equilibrated to 10% B for 10 min before subsequent injections.

For MALDI TOF/TOF (tandem time-of-flight mass spectrometry (MS/MS) mass spectrometry, the trypsin digested (Trypsin Gold, Madison, WI, USA) protein samples were analyzed with an AB Sciex TOF/TOF 5800 System (SCIEX, Darmstadt, Germany) instrument operated in the positive ion mode [24]. The MALDI stage was set to continuous motion mode. Mass spectrometry (MS) data were acquired at 2400 laser intensity with 1000 laser shots/spectrum (200 laser shots/sub-spectrum), and tandem mass spectrometry (MS/MS) data were acquired at 3300 laser intensity with a DynamicExit algorithm and a high spectral quality threshold or a maximum of 1000 laser shots/spectrum (250 laser shots/sub-spectrum). Up to 30 top precursors with S/N > 30 in the mass range 800–4000 Da were selected from each spot for MS/MS analysis. Mass spectrometry analysis was performed using the TOF/TOF Series Explorer software. MS and MS/MS spectra were analyzed using specialized software ProteinPilot 4.0 (AB Sciex) [25] in the MASCOT search engine (or Protein Prospector [26]) based on the international protein databases UniProtKB/Swiss-Prot/NCBI [27]. Carbamidomethyl cysteine was set as a fixed modification. False discovery rate (FDR) analysis was done by analysis of reversed sequences using the embedded PSEP tool. The exponentially modified protein abundance index (emPAI index) was calculated using the following equation:emPAI = 10^(Nobserved/Nobservable)^ − 1,(1)
where the number of experimentally identified peptides (N_observed_) and the number of theoretically possible peptides for this protein (N_observable_).

This information indirectly indicates the localization of the protein and changes its expression in the semi-quantitative assessment, and it can be used to guide the differences in cellular behavior in the control and experimental groups. We used a normalized emPAI value calculated for each protein as a fraction of the maximum detected index value in each sample [28].

## 3. Results and Discussion

Brewing, baking, and ethanol production are the main technologies where *S. cerevisiae* yeast plays a major role. For a comparative proteome study of yeast, adapted to different technological processes, we used brewer’s Y-3194, baker’s LV7, and ethanol-producing ER strains. To obtain comparative data on the composition of the yeast proteome, proteins were separated in a two-dimensional electrophoresis system (Figure 2).

To determine differences in the expression levels of certain proteins, we scanned and compared corresponding spots in colored gels. Figure 2 shows the results of two-dimension electrophoresis of total yeast lysates for three studied strains.

Separation with two-dimensional high-resolution electrophoresis provided a possibility to observe yeast cell lysates protein profiles, differing in the quantitative and qualitative composition of the spots. The images of two-dimensional electrophoresis were visually analyzed. The yeast strains total lysates were digested with modified trypsin, and subsequent sample preparation for LC-MALDI TOF/TOF mass spectrometry was performed.

To clarify the relative abundance of proteins in the samples, the emPAI index was used; it was calculated for analysis in the Mascot search [28]. This approach involves semi-quantitative estimation of the relative protein content of the mixture based on sequence coverage and database search results. Unlike other approaches, this method requires no additional data (in addition to MS/MS mass spectra) and needs no adjustments of the mass spectrometry analysis parameters. The emPAI was calculated using Equation (1) (Materials and Methods).

The results of mass spectrometry analysis allowed us to identify the number of proteins that differed among the samples (Figure 3).

To evaluate changes in the content of certain proteins in yeast lysates at the semi-quantitative level, we used the distribution of reliably identified proteins of each strain according to the values of logarithms of emPAI indexes.

Thus, the mass spectrometry analysis made it possible to identify proteins with increased or, conversely, reduced expression. The emPAI index clarified the relative protein content in the samples. The comparison of the protein content in different strains demonstrated that approximately one-third of all cell proteins from baker’s and brewer’s strains possessed the same expression, whereas in the ethanol-producing strain, only 23% of the proteins were synthesized in the same amount (Table 1, Figure 4).

Twenty-one per cent of the protein content of brewer’s yeast Y-3194 showed an increased expression in respect to the same proteins of baker’s strain LV7, 49% of proteins decreased, and 30% of proteins had the same expression (Figure 4a). Sixty-one per cent of proteins of ethanol-producing strain ER increased, 16% decreased, and 23% were unchanged in respect to Y-3194 (Figure 4b). Forty per cent of proteins showed an increased expression in ER, 37% decreased, and 23% were unchanged in respect to baker’s strain LV7 proteins (Figure 4c).

Brewer’s and baker’s strains reacted approximately the same under identical conditions of the experiment, while the ethanol-producing strain had an increased number of proteins with altered (increased or decreased) expression. Thus, the proteomes of baker’s and brewer’s yeast are more similar. This fact correlates with genome analysis evolutionary data, showing that baker’s and brewer’s industrial yeast strains are more similar in terms of the evolutionary aspect (Table 1).

Table 2 shows data on specific proteins that have increased expression in the brewer’s strain Y-3194, baker’s strain LV7, and ethanol-producing strain ER.

The proteomic sets of investigated strains clearly showed that particular proteins had an increased expression, typical for each strain. In Table 2, we present the proteins that had 1.36- to 15.5-fold induction in one strain in respect to another. We collected information on such proteins, corresponding genes, and their role in yeast metabolism and tried to find the link between their function and perspectives for domestication. Metabolic reaction specificity and target genes for domestication in industrial yeast are not the same for brewing, baking, and ethanol production because of the medium content and production stresses. To overcome specific stresses and efficient production of the needed final products with high quality, industrial yeast should have corresponding genes and regulation elements in their genome. This is their potential for further domestication. Such genes could be targets for new industrial yeast construction directed for particular industry. For example, if a strain has POF+ phenotype, it is not a good candidate for brewing until corresponding genes are functional. NADP-specific glutamate dehydrogenase GDH3 gene expression is induced by ethanol and suppressed by glucose, therefore, high expression of this gene could give an advantage for ethanol-producing strains. The domestication perspectives of most yeast genes is not known. Our proteomic study detected some genes with domestication potential.

Regulation of ribosomal protein synthesis depends on the composition of nutrient factors and the corresponding regulation of a number of signaling pathways, which can dramatically induce or inhibit transcription of ribosomal genes, and lead to serious consequences for the expression of other genes [29].

In comparison with the baker’s strain LV7, the content of Rps19p (encoded by *RPS19A* gene) and Rps14p (encoded by *RPS14A* gene) 40S ribosomal subunit proteins in brewer’s yeast Y-3194 increased by 11.3- and 4.49-fold (Table 2).

Rps19 is an essential protein necessary for the biogenesis of the small ribosome subunit. Disorder of the yeast *RPS19* genes causes a decrease in the growth rate and affects the formation of 40S subunits. Mutations in ribosomal protein S19 in humans are associated with Blackfan anemia, which usually manifests itself in early infancy and is accompanied by craniofacial abnormalities, lack of growth, predisposition to cancer, and other congenital abnormalities [30].

The difference between the strains affects not only ribosome biogenesis, but also the expression of *HHF1* (encoding histone 4) and *RVB1* (expressing ATP-dependent DNA-helicase) genes responsible for chromatin packaging. Their expression increased 9.28- and 2.18-fold, respectively (Table 2).

Chromatin consists of nucleosomes, and each nucleosome contains an octamer formed by two copies of histones H2A-H2B and H3-H4 heterodimers. The location of nucleosomes along chromatin is involved in the regulation of gene expression, since the packing of DNA into nucleosomes affects the availability of transcription initiation sequences. Nucleosomes prevent many DNA-binding proteins from approaching their sites [31], whereas correctly positioned nucleosomes can bring non-adjacent DNA sequences into close proximity, promoting transcription. *RVB1* encodes the ATP-dependent DNA-helicase subunit of the Ino80 nuclear complex, which is involved in the cell cycle, chromatin remodeling, and transcription regulation [32].

Two genes associated with phosphate metabolism, PHO11 and IPP1, also showed an increased expression in brewer’s yeast (3.27- and 1.46-fold, respectively, Table 2). Yeast phosphate-repressed acid phosphatase is an extracellular enzyme encoded by three structural genes, i.e., *PH05* (p60), *PHO10* (p58), and *PHO11* (p56) [33]. Inorganic pyrophosphatase encoded by *IPP1* gene is an important enzyme that plays a key role in a wide range of cellular biosynthetic reactions, such as the synthesis of amino acids, nucleotides, polysaccharides, and fatty acids [34].

Particular attention was paid to the increased expression of alcohol dehydrogenase 4 encoded by the *ADH4* gene in brewer’s yeast Y-3194. This fact is in agreement with earlier studies that reported an overexpression of *ADH4* gene in yeast used in brewing [35]. *S. cerevisiae* has five genes encoding alcohol dehydrogenases involved in ethanol metabolism, i.e., *ADH1*, *ADH2*, *ADH3*, *ADH4*, and *ADH5*. Four of the enzymes encoded by these genes, i.e., Adh1p, Adh3p, Adh4p, and Adh5p, reduce acetaldehyde to ethanol during glucose fermentation, while Adh2p catalyzes the reverse reaction of ethanol oxidation to acetaldehyde. All five alcohol dehydrogenases and the Sfa1p enzyme are also involved in the production of fusel alcohols during fermentation [35]. Fusel alcohols are the end products of amino acids catabolism (valine, leucine, isoleucine, methionine, phenylalanine, tryptophan, and tyrosine) along the Ehrlich pathway and contribute to taste and aroma of fermented yeast foods and beverages [36]. A decrease in the zinc content in the medium induces *ADH4* transcription [37]. Two genes, *ADH4* and *ZRT1,* with increased expression in brewer’s yeast are associated with zinc, according to our proteomic data, and reflect a well-known fact that zinc concentration is very important for the beer fermentation process (Table 2). Noteworthy, *ZRT1* is a gene encoding a high-affinity zinc transporter [38].

Another gene, *COF1,* with an increased expression in brewer’s yeast strain Y-3194, which is involved in the processes of secretion and proteins sorting, is known to be of great importance in the metabolism of brewer’s yeast. Thus, it has been revealed that there was an increase in the expression of the *COF1* gene, which encoded cofilin. Cofilin is involved in pH-dependent depolarization of actin filaments and participates in selective sorting and export of secretory cargo from the Golgi apparatus structures [39].

An interesting fact is a small increase in the expression of two dehydrogenases in brewer’s yeast, i.e., peroxisomal malate dehydrogenase encoded by *MDH3* gene and NAD-specific glutamate dehydrogenase (encoded by *GDH2* gene). Peroxisomal malate dehydrogenase catalyzes the mutual conversion of malate and oxaloacetate participates in the glyoxylate cycle [40]. NADP (+)-dependent glutamate dehydrogenase degrades glutamate to ammonia and alpha-ketoglutarate [41].

The expression of the *RPL8A* and *HOM2* genes encoding the ribosomal protein 60S L8-A and aspartic beta-semialdehyde dehydrogenase, respectively, was increased more than five times in the baker’s yeast strain LV7 as compared with the brewer’s yeast strain.

The L8A 60S protein of the ribosome subunit is necessary for the conversion of 27SA3 pre-rRNA to 27SB pre-rRNA during the assembly of a large ribosomal subunit [42]. Another 40S ribosomal protein, S26-Bp, with increased expression was found [43].

Aspartic beta-semialdehyde dehydrogenase catalyzes the second stage of the general pathway of methionine and threonine biosynthesis, regulated by the Gcn4p protein, which provides general control of amino acid synthesis [44]. *GCN4* encodes a transcriptional activator during amino acid starvation, participates in 19 out of the 20 pathways of amino acid biosynthesis, and Gcn4p can directly or indirectly regulate the expression of genes involved in purine biosynthesis, organelle biosynthesis, glycogen homeostasis, and multiple stress reactions [45].

In addition, the expression of Ade13p adenylosuccinate lyase (encoded by *ADE13* gene), which catalyzes two stages of the purine nucleotide biosynthesis pathway, was increased by almost five times in the LV7 yeast strain. The first reaction leads to the formation of inosine monophosphate (IMP), and the second one converts IMP to adenosine monophosphate (AMP) [46]. Ade13p is a widely conserved protein, and its orthologs have been described in bacteria and humans. Purine derivatives play an important role in the chemistry of natural compounds, such as the purine bases of DNA and RNA, coenzyme NAD, alkaloids, caffeine, theophylline, and theobromine; therefore, they are widely used in pharmaceuticals.

The *RGI1* gene (induced respiratory growth protein) is involved in energy exchange under respiratory conditions, and its expression increases during intracellular iron depletion or in response to DNA replication stress [47]. Interestingly, the baker’s yeast increased the expression of *GRE1* (GRE1p protein). Previously, its expression was shown to be increased under osmotic, oxidative, thermal, and chemical stress in yeast. This may be necessary for baker’s yeast to overcome osmotic stress. It is regulated by the mitogen-activated protein kinase Hogp, which is involved in osmoregulation [48]. Protein kinases regulate the cell cycle, differentiation, and metabolic pathways. Disturbance of these processes can cause apoptosis, i.e., programmed cell death.

The expression of the ERG9 gene encoding farnesyl diphosphate-farnesyltransferase (also called squalene synthase) was increased by more than three times in the baker’s yeast strain [49]. This enzyme connects two fragments of farnesyl pyrophosphate to form squalene in sterol biosynthesis pathway. In the human body, squalene acts as an antimicrobial, anti-carcinogenic, and fungicidal agent. It is possible that the protective properties of squalene allow baker’s yeast to overcome osmotic stress in the process of dough forming.

A three-fold increase was found in the expression of the *SEC23* gene encoding GTPase-activating protein, a component of the Sec23p-Sec24p heterodimer of the COPII vesicle envelope involved in endoplasmic reticulum (ER) transport to Golgi. COPII is a coatomer, which is a type of vesicle envelope protein that transports proteins from the endoplasmic reticulum to the Golgi apparatus [50]. The COPII vesicle coat minimally consists of the following five subunits: the Sar1p GTPase, the Sec23p-Sec24p heterodimer, and the Sec13p-Sec31p complex. The formation of COPII vesicles requires assembly of the COPII vesicle coat and cargo sorting and is regulated by GTP hydrolysis cycles. Sec24p participates in cargo sorting [51].

Thus, we have identified a unique feature of the baker’s yeast proteome to increase the expression of proteins involved in the reactions of the yeast cell to overcome stressful conditions.

The main gene, in which regulation in the ethanol-producing yeast strain ER is 15.5-fold increased as compared with the brewer’s yeast Y-3194, is the *GDH3* gene encoding NADP (+)-dependent glutamate dehydrogenase, which synthesizes glutamate from ammonia and alpha-ketoglutarate. Its expression is regulated by nitrogen and carbon sources. *GDH3* expression is induced by ethanol and suppressed by glucose. Under conditions of carbon source deficiency, Gdh3p is a key isoform involved in alpha-ketoglutarate distribution for glutamate biosynthesis and energy metabolism [52]. The absorption of ammonium by the cell is closely related to such fundamental cellular processes as the synthesis of amino acids. NADP-dependent glutamate dehydrogenase Gdh3p has a pleiotropic effect and is involved in chromatin regulation, nitrogen catabolite repression, actin cytoskeleton, and apoptosis. Glutamate plays an important role in maintaining the redox potential of the cell.

During metabolism, the toxic molecules, methylglyoxal and glyoxal, are synthesized. Detoxification of these molecules is a glutathione-dependent system with glyoxalase Glo1p, which catalyzes the detoxification of methylglyoxal (a byproduct of glycolysis) by condensation with glutathione to form S-D-lactoylglutathione. The expression is regulated by methylglyoxal levels and osmotic stress. We observed an increase in glyoxalase expression by 3.67-fold in the ethanol-producing yeast strain ER as compared to the brewer’s yeast strain. This is probably due to the more severe conditions of ethanol stress experienced by ethanol-producing yeast.

A similar increase in the expression of the *HEM2* gene encoding delta-aminolevulinic acid dehydrogenase, which catalyzes the conversion of 5-aminolevulinate to porphobilinogen, the second stage of heme biosynthesis, was observed. Heme consists of a porphyrin ring (formed by four pyrrole rings) and a Fe^2+^ ion. Due to changes in the oxidation degree of the iron ion, electrons move along the electron transport chain, which ensures the oxidation of various substrates during respiration [53].

In ethanol-producing yeast, the expression of the *GDB1* gene is increased. The glycogen cleavage enzyme has glucanotransferase and alpha-1,6-amyloglucosidase activity, and protein expression increases in response to DNA replication [54]. *GDB1* expression is induced in the late exponential growth phase and in response to various stresses [45]. An increased expression of two genes of the large ribosome subunit *RPL24B* (encodes 60S ribosomal protein L24) and *RPL4A* (encodes *60S* ribosomal protein L4) was detected.

The content of the D subunit of the V-ATPase peripheral membrane domain V1, part of the electrogenic proton pump found throughout the endomembrane system, is increased in ER yeast strain and plays a role in connecting proton transport and ATP hydrolysis. *VMA8* encodes the D subunit of the yeast V-ATPase V1 domain [55]. The ATP-dependent proton pumps acidify intracellular vacuolar compartments, which is important for many cellular processes, including endocytosis, targeting of lysosomal enzymes, and other molecular processes. It is possible that the activity of the *VMA8* is important for enhancing the processes of secretion and overcoming ethanol stress in ethanol-producing yeast strains.

Ethanol-producing yeast has an increased content of S-adenosyl-L-methionine-dependent tRNA, m5C-methyltransferase (encoded by *NCL1* gene), which methylates cytosine in several positions in tRNA and intron-containing pre-tRNAs and causes selective translation of mRNA genes enriched with the TTG codon. Mutations in the *NCL1* gene cause hypersensitivity to oxidative stress.

Thus, most of the identified genes, which expression is increased in ethanol-producing yeast strain as compared with brewer’s yeast, relate to redox processes, secretion, and stress surviving, which could give an advantage in toxic conditions of high ethanol concentrations.

In ethanol-producing yeast as compared with baker’s yeast, the expression of NADHX epimerase (encoded by *NNR1* gene) associated with DNA repair and the function of proteasomes [56], and the translation initiation factor TIF45p [57], is sharply increased, which may indicate an increased rate of energy-dependent processes and protein synthesis in the yeast cell. Under conditions of high ethanol concentrations, which lead to changes in status of redox potential of the cell, NADH accumulates due to the hydrolysis of ethanol by alcohol dehydrogenases, which leads to the accumulation of acetaldehyde. This change has a significant effect on the metabolic reactions in the cell, as it provides an advantage to the synthesis and accumulation of fatty acids, preventing gluconeogenesis and inhibits the tricarboxylic acid cycle.

Overexpressed in ethanol-producing yeast *PMI40* encodes phosphomannose isomerase, which catalyzes isomerization between mannose-6-phosphate (M6P) and fructose-6-phosphate. M6P is a substrate for Sec53p and is a source of mannose for N- and O-bound glycosylation of proteins and their GPI fixation on the plasma membrane. The removal or thermal inactivation of Pmi40p leads to abnormal morphology, defective secretion, and the appearance of glycoproteins on the cell surface [58]. Effective cellular transport and the correct localization of proteins provide the advantage of ethanol-producing yeast strains.

Zinc metalloendopeptidase cleaves the leader sequence of proteins imported into mitochondria. Prd1p expression increases in response to DNA replication stress [59]. Metallopeptidase or Prd1p releases mitochondria from misfolded proteins.

Under ethanol stress, Hpt1p (hypoxanthine-guanine phosphoribosyltransferase, encoded by *HTP1* gene) catalyzes the synthesis of purine nucleoside monophosphates, which is important for creating a pool of nucleotides from derivatives in the cell to ensure efficient synthesis of nucleic acids and potential accumulation of nucleoside triphosphates, which are energy accumulators [60].

Cysteine proteinase Rpt5p (regulatory subunit of 26S proteasome, encoded by *RPT5* gene) may be involved in rapid changes in protein composition under toxic conditions of ethanol stress. Proteasome ATPase is involved in the degradation of polyubiquitinated substrates [61].

Mitochondrial function is particularly important in conditions of oxidative stress in a high alcohol environment. That may explain why the ethanol-producing yeast strain ER has an increased expression of mitochondrial cytochrome C peroxidase encoded by the *CCP1* gene. Mitochondrial cytochrome c peroxidase degrades reactive oxygen species in mitochondria involved in the reaction to oxidative stress [62].

The expression of the high-affinity hexose transporter Hxt6p (encoded by *HXT6* gene) [63] and alpha-glucosidase Mal32p (encoded by *MAL32*) [64] was increased in the ER strain. Efficient transport and sugars uptake are undoubtedly important for survival in toxic environments. The screening also revealed genes encoding ribosomal proteins and proteins involved in secretion and cargo distribution inside the cell, as well as a number of genes with unknown function. Possibly, certain technological adaptations of strains require the interaction of certain ribosome proteins to ensure effective transcription under certain conditions. Effective secretory activity creates technological advantages for strains. Thus, the identification of genes with a previously unknown function provides information about the potential role of these genes in osmotic and ethanol stress.

## 4. Conclusions

Domestication of microbes has both a historical and practical value. Whole-genome sequencing efforts and proteomics facilities allows researchers to track, compare, and reproduce the route to microbial domestication, as well as practically apply this knowledge. However, some genome changes remain unclear. On the one hand, domesticated microbes own many advantages due to their higher fermentation rates and more balanced and consistent aroma profiles. On the other hand, availability of whole-genome sequencing data, combined with proteomics analytical approaches allows researchers to generate modern industrial strains by using more up-to-date technologies for introducing targeted gene modifications. The essential challenge in new industrial yeast selection is the lack of selective markers, mainly for creating aroma profile, which is important for consumers. The use of classical methods of genetics and selection is labor intensive and the result does not always satisfy modern industry, since recessive alleles of the desired genes cannot express after hybridization. Moreover, there is a chance of developing new undesirable phenotypic properties.

Our proteomic analysis revealed genes of large and small ribosomal subunits proteins. This suggests a possible innovative link between specific ribosomes synthesis and certain conditions, such as ethanol or osmotic stress. In addition, we found genes with unknown function, *YPR127W* and *YNL274C*, with increased expression in ethanol-producing strains. Probably, the function of those genes is connected with yeast adaptation to toxic conditions during ethanol production.

Proteomic analysis of brewer’s yeast showed an overexpression of zinc-dependent alcohol dehydrogenase 4 and high-affinity zinc-regulated transporter 1, reflecting the well-known fact that zinc is important for beer fermentation process.

Yeast adapts to osmotic stress during dough formation and maturation. In our study, baker’s yeast demonstrated increased amount of GRE1p protein, involved in osmoregulation. Previously, the expression of this gene was shown to be increased under osmotic, oxidative, thermal, and chemical stress in yeast. We assume that overexpression of *GRE1* is necessary for baking yeast to overcome osmotic stress.

In conditions of ethanol production, yeast should be tolerant to toxic stress, caused by high ethanol concentrations. In our screening, we found that ethanol-producing yeast strain had an increased amount of NADHX epimerase, which was involved in repairing nicotinamide nucleotides. This may indicate an increased rate of energy-dependent processes.

The data obtained is important in the future for the targeted construction of appropriate production strains with improved properties applying modern technologies for gene expression modification. Our results provide the background for further application of the genome editing technology based on the CRISPR-Cas9 as an already established tool for modifying the phenotypes of industrial yeasts and correcting their possible undesirable properties.

## Figures and Tables

**Figure 1 bioengineering-07-00147-f001:**
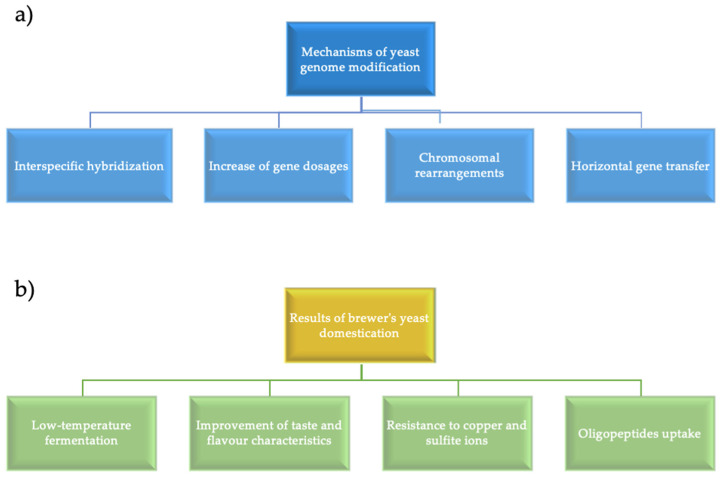
(**a**) Mechanisms of genome evolution; (**b**) Results of brewer’s yeast domestication [1].

**Figure 2 bioengineering-07-00147-f002:**
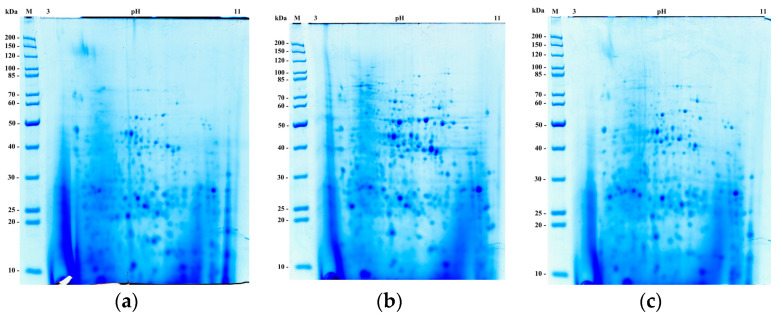
Two-dimensional electrophoresis of total yeast lysates from the strains. (**a**) Y-3194; (**b**) LV7; (**c**) Ethanol Red (ER). On the left, molecular weight markers, on the top, isoelectric point values (pH/pI).

**Figure 3 bioengineering-07-00147-f003:**
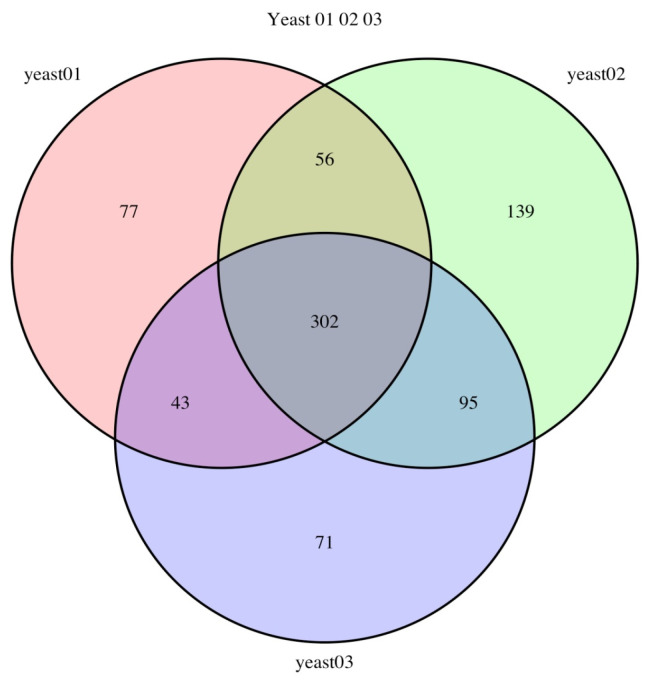
Venn diagram illustrating the distribution of proteins reliably identified by tandem mass spectrometry from total yeast lysates of Y-3194, LV7, and ER 1–3 strains (yeast 01, yeast 02, and yeast 03, respectively).

**Figure 4 bioengineering-07-00147-f004:**
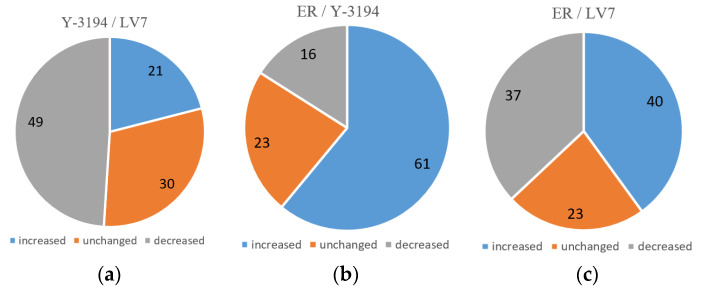
Comparison of proteomes of production strains (relative number of proteins). (**a**) Y-3194 proteins in respect to LV7; (**b**) ER proteins in respect to Y-3194; (**c**) ER proteins in respect to LV7.

**Table 1 bioengineering-07-00147-t001:** Data on the analysis of the relative protein content in the proteomes of brewer’s Y-3194, baker’s LV7, and ethanol-producing ER strains.

Protein Expression	Quantity of Proteins					
	Y-3194/LV7		ER/Y-3194		ER/LV7	
	n	%	n	%	n	%
Induced	12	21	27	61	38	40
Not changed	17	30	10	23	22	23
Decreased	28	49	7	16	35	37
Sum	57	100	44	100	95	100

**Table 2 bioengineering-07-00147-t002:** Genes with increased expression in the studied yeast strains.

Gene	Protein	Protein Expression Induced, Fold
	The Brewer’s Strain Y-3194 in Relation to the Baker’s Strain LV7	
*RPS19A*	40S ribosomal protein	11.3
*HHF1*	Histone H4	9.28
*RPS14A*	40S ribosomal protein	4.49
*PHO11*	Acid phosphatase PHO11	3.27
*ADH4*	Alcohol dehydrogenase 4	3.25
*RVB1*	Ru VB-like protein 1	2.18
*ZRT1*	Zinc-regulated transporter 1	2.07
*COF1*	Cofilin OS	1.80
*MDH3*	Malate dehydrogenase, peroxisomal	1.62
*IPP1*	Inorganic pyrophosphatase	1.46
*GDH2*	NAD-specific glutamate dehydrogenase	1.36
	The Baker’s Strain LV7 in Relation to the Brewer’s Strain Y-3194	
*RPL8A*	60S ribosomal protein L8-A p	5.18
*HOM2*	Aspartic beta-semialdehyde dehydrogenase	5.14
*ADE13*	Aspartic beta-aldehyde dehydrogenase	4.90
*RPS26B*	40S ribosomal protein S26-Bp	4.69
*RGI1/YER067W*	Induced Respiratory Growth	4.16
*GRE1*	GRE1p protein	4.06
*YCP4*	Flavoprotein-like protein YCP4p	3.54
*ERG9*	Squalene synthase	3.36
*SEC23*	Transport protein SEC23p	3.00
	The Ethanol-Producing Strain ER in Relation to the Brewer’s Y-3194	
*GDH3*	NADP-specific glutamate dehydrogenase	15.5
*GLO1*	Lactoylglutathione lyase	3.67
*HEM2*	The glycogen-splitting enzyme	3.60
*GDB1*	60 Ribosomal protein L24-B	3.33
*RPL24B*	60S ribosomal protein L24	2.39
*VMA8*	Type V proton ATPase D Subunit	2.30
*RPL4A*	60S ribosomal protein L4	2.27
*YPR127W*	Putative pyridoxal reductase	2.20
*YNL274C*	Putative 2-hydroxy acid dehydrogenase	2.20
*NCL1*	RNA (cytosine-5-)-methyltransferase	2.14
	The Ethanol-Producing Strain ER in Relation to the Baker’s LV7	
*NNR1*	NADHX epimerase	10.36
*TIF45*	Eukaryotic translation initiation factor 4E	5.92
*PMI40*	Mannose-6-phosphate isomerase	4.83
*PRD1*	Zinc metalloendopeptidase	4.43
*HPT1*	Hypoxanthine-guanine phosphoribosyltransferase	3.83
*CCP1*	Cytochrome C peroxidase, mitochondrial	3.64
unknown	Uncharacterized vacuole membrane protein	3.13
*RPT5*	Regulatory subunit of 26S proteasome	3.08
*HXT6*	High affinity hexose Transporter	2.38
*MAL32*	Alpha-glucosidase	2.35

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
