# Peer review of "Proteomics Answers Which Yeast Genes Are Specific for Baking, Brewing, and Ethanol Production"

_bioengineering, 2020, doi:10.3390/bioengineering7040147_

Round 1
Reviewer 1 Report
In general, the paper entitled: “Proteomics answers which yeast genes are specific for bakery, brewing and ethanol production”, describes experiments of a good quality with relevant analyses. The authors of the manuscript presented a detailed analysis of the protein profiles of baker's, brewing and distilling yeasts. The manuscript showed differences in the activity of genes responsible for coding metabolic proteins in different strains of S. cerevisiae. The reported study is worthy of investigation, and might be very interesting for the research in this field. The publication is recommended, subjected to minor revision.
However, here are few points which must be considered during revision of the manuscript:
#1 Indicate the innovative nature of the research. Demonstrate the difference of the presented research results in relation to the research of other authors.
#2 It seems reasonable to summarize the potential practical application of the research results obtained.
Author Response
Thank you for your time and patience during revision of our manuscript.
Here are our notes.
Point 1: indicate the innovative nature of the research. Demonstrate the difference of the presented research results in relation to the research of other authors.
We indicated the innovative nature of the research in Summary (P.1 L21-22) and P.3 L.211-233.
Point 2: it seems reasonable to summarize the potential practical application of the research results obtained.
Response 2: we marked and summarized the potential practical application of the research results obtained in Abstract and Conclusions
P.1 L25 and P.15-16 L643-767
Thank you.
Reviewer 2 Report
Title: Proteomics answers which yeast genes are specific for bakery, brewing and ethanol production Authors: Svetlana Davydenko , Tatiana Meledina , Aleksey Mittenberg , Sergey Shabelnikov , Maksim Vonsky , Artyom Morozov General comments: The manuscript is a comparative trial based on a proteomics study of the three strains of Saccharomyces cerevisiae used in the food industry.The authors make an attempt to compare individual proteins encoded by selected genes to illustrate the differences between strains.
The goal is interesting, but the execution has important drawbacks:
1) the authors compare only 3 strains and this is not enough to come to the intended conclusions that will allow targeted improvement of industrial strains construction (abstract),
2) why only 3 strains were used. What was their choice (there are many strains for use in distilling, brewing and baking)
2) the results are not shown clearly. Instead of Tables 2-5 comparing strain against strain, a one table showing the characteristics of all strains would be indicated.
3) The conclusion should be contained in a separate chapter. In its current form, clonclusion does not answer the purpose indicated in the abstract: "The data obtained would allow targeted improvement of industrial strains construction"
4) The authors cite over 40% of old literature, more than 15 years old. Proteomics research should be based on the latest data in this field. The authors also cite their articles not directly related to the topic of this manuscript, e.g. items 17 and 19.
Detailed comments:
Pg 1, L38: pharmaceuticals
Pg 4, L119: Were the suspensions standardized? how?
Pg 4, L114: S. cerevisiae must be italicized
Pg 4, L147: Results and Discussion
Pg 5, Figure 2: Photos are not clearly visible.
Pg 6, Figure 3: The diagram does not reflect the results, it is not indicated here
Pg 6, Figure 4: Little visible results
Author Response
Thank you for your time and patience during the revision of our paper.
Here are our notes.
Point 1: the authors compare only 3 strains and this is not enough to come to the intended conclusions that will allow targeted improvement of industrial strains construction (abstract).
Response 1: those industrial yeast strains were selected as representatives of different fermentation processes because we had the experience, using them for beer, bread, and distillation.
Point 2: why only 3 strains were used. What was their choice (there are many strains for use in distilling, brewing, and baking)
Response 2: you are right that there are a lot of industrial strains all over the world but many of them are not available because they are commercial strains with certain limitations for using them. On the other hand, as an expert, you surely realize that proteomic analysis is very complicated, time, and resource-consuming, and it is not possible to analyze hundreds of existing industrial strains. This work should be done step by step. Our research is only an example of the general approach that we present. More data from other scientists should be compared.
Point 3: the results are not shown clearly. Instead of Tables 2-5 comparing strain against strain, one table showing the characteristics of all strains would be indicated.
Response 3: to clarify results we merged Tables 2-5 in one Table 2 (P.7 L380).
Point 4: the conclusion should be contained in a separate chapter. In its current form, conclusion does not answer the purpose indicated in the abstract: "The data obtained would allow targeted improvement of industrial strains construction"
Response 4:we arranged a separate chapter for Conclusions, explaining how obtained data could allow targeted improvement of industrial strains, showing the specific target genes for brewing, baking, and distilling yeast. P.15-16 L643-767.
Point 5: the authors cite over 40% of old literature, more than 15 years old. Proteomics research should be based on the latest data in this field. The authors also cite their articles not directly related to the topic of this manuscript, e.g. items 17 and 19.
Response 5: Papers that describe genomic and proteomic approaches [3, 4, 5, 6, 11, 13] are published in 2017-2019. CRISPR-Cas papers [7, 8, 9, 10, 12] from 2014-2019. Other papers are connected with certain genes and their function. Most of them were based on classical genetics and molecular biology methods. Our approach was to mention the original papers because it is more convenient to the readers. This reflects how historically yeast genetics was developed. Other possibility is to substitute all such references with the reference to SGD. Then our readers have to look for each of the mentioned genes by themselves, which is probably not so convenient to them.
In items 17, 19 we refer to the methods that are more detailed there. We followed the same protocol
Detailed comments:
1. Pg 1, L39: pharmaceuticals
P.1 L39 pharmaceutical changed to pharmaceuticals
2. Pg 4, L119: Were the suspensions standardized? how?
P.4 L249-252 We added details about yeast standardization:
Yeast cultivation. Yeast cultures were grown on a solid YPD medium for 24 h at 27 °C for the stationary phase, then inoculated for standardization at initial concentration 106 cells/ml in flasks with 100 ml of liquid YPD and grown under constant agitation at 150 rpm for 18 h at 27 °C. Yeast cells were cultivated simultaneously in the same YPD batch.
3. Pg 4, L114: S. cerevisiae must be italicized
P.4 L244 S. cerevisiae changed to S. cerevisiae
4. Pg 4, L147: Results and Discussion
P.5 L314 Results changed to Results and Discussion
5. Pg 5, Figure 2: Photos are not clearly visible.
We apologize that our original Figure 2 is not clearly visible in the manuscript, but there is an option to enlarge figures in the on-line version of the manuscript.
5. Pg 6, Figure 3: The diagram does not reflect the results, it is not indicated here
Figure 3 The General picture of protein intersections by their qualitative representation is shown using Venn diagrams and graphs reflecting the occurrence of proteins from total lysates (Figures 3).
7. Pg 6, Figure 4: Little visible results
We have decided to avoid attaching figure 4 due to the lack of information it represents
Additionally, we did changes in English throughout the whole manuscript as recommended.
Thank you.
Round 2
Reviewer 2 Report
Dear Authors
The manuscript may may be published in its current form. This version is significantly improved.
One bad point: English should be improved
Author Response
Dear Reviewer,
thank you for reviewing the manuscript.
During the second revision, we did changes in restructuration, grammatical errors, typos, delete duplication throughout the whole manuscript.
Thank you.